# Stribeck Curve of Magnetorheological Fluid within Pin-on-Disc Configuration: An Experimental Investigation

**DOI:** 10.3390/ma13204670

**Published:** 2020-10-20

**Authors:** Jakub Roupec, Filip Jeniš, Zbyněk Strecker, Michal Kubík, Ondřej Macháček

**Affiliations:** Faculty of Mechanical Engineering, Brno University of Technology, Technická 2, 616 69 Brno, Czech Republic; roupec.j@fme.vutbr.cz (J.R.); filip.Jenis@vutbr.cz (F.J.); strecker@fme.vutbr.cz (Z.S.); ondrej.Machacek@vutbr.cz (O.M.)

**Keywords:** friction, Stribeck curve, magnetorheological fluid, pin on disc, MR damper, tribological performance

## Abstract

The paper focuses on the coefficient of friction (COF) of a magnetorheological fluid (MRF) in the wide range of working conditions across all the lubrication regimes—boundary, mixed, elastohydrodynamic (EHD), and hydrodynamic (HD) lubrication, specifically focused on the common working area of MR damper. The coefficient of friction was measured for MR fluids from Lord company with concentrations of 22, 32, and 40 vol. % of iron particles at temperatures 40 and 80 °C. The results were compared with a reference fluid, a synthetic liquid hydrocarbon PAO4 used as a carrier fluid of MRF. The results show that at boundary regime and temperature 40 °C all the fluids exhibit similar COF of 0.11–0.13. Differences can be found in the EHD regime, where the MR fluid COF is significantly higher (0.08) in comparison with PAO4 (0.04). The COF of MR fluid in the HD regime rose very steeply in comparison with PAO4. The effect of particle concentration is significant in the HD regime.

## 1. Introduction

Magnetorheological (MR) fluid belongs into the group of smart materials enabling a change in an apparent viscosity in over a great range, and this switching between two states can be reached within 1 ms [1]. The fluid is the suspension of micron-sized particles, which are usually made of pure iron because of their favorable magnetic properties. These particles have a globular shape, and they are suspended in the carrier fluid, for instance, water, silicone oil, or, in most cases, a synthetic hydrocarbon, say, polyalphaolefin (PAO). When the magnetic field is applied, the ferromagnetic particles are concatenated into chain formation along the magnetic flux lines. The cohesion of chain formations causes an increase in the apparent viscosity of MR fluid [2]. This property is most often exploited by MR dampers [3,4,5], brakes/clutches [6,7,8], seals [9,10], or flexible structures [11,12]. The applicability of these MR devices in smart mechanical systems has several limitations. The main issues are as follows: (i) the sedimentation stability [13,14]; (ii) an increase of MR fluid consistency during the MR fluid loading called an in-use-thickening [15,16]; (iii) and the abrasiveness and poor lubricating properties of the MR fluid [17]. Currently, the first two issues are being sufficiently solved by using suitable additives to prevent the sedimentation and oxidation of iron particles.

The MR device lifetime is limited by two major sources of wear: (i) fluid flow wear and (ii) wear inside the tribological contact pairs by iron particles or additives. Wear due to fluid flow in the damper has been described in, e.g., [18]. Wear causes a decrease in the required force or torque due to an increase in the gap size. However, this problem can be solved by a suitable damper control strategy. The wear or higher friction losses of tribological contact pairs is a significant problem for MR device lifetime and performance. In the case of the MR damper, several tribological contact pairs can be found: sealing and guiding of (i) the floating piston, (ii) the piston, and (iii) the piston rod. Wear of these parts can cause worse functionality exhibited as a higher friction force and MR fluid leakage from the device or into the gas chamber in the damper. Therefore, the basic knowledge of wear and lubricating behavior of MR fluid in different tribological contact pairs can significantly help in practice. Jolly [19] measured the coefficients of friction for MR fluid lubricated iron-on-iron conformal interfaces. He compared the four types of MR fluid from Lord with dry friction (0.18), and the measured coefficient of friction (COF) was an interval from 0.04 to 0.07 for all samples. Song [20] tested material pairs of steel-on-steel and aluminum-on-aluminum. The normal force was applied at three levels and at three levels of speed rotation. The COF was lower for the highest speed and the highest load. The lowest COF was measured for the steel-on-steel configuration and the highest COF for the aluminum material pair. Shahrivar [21] measured Stribeck curves of ferrofluids and MR fluid MRF-132DG from LORD. He used configuration ball-on-three plates from stainless steel AIS 316. The COF was increased from 0.13 to 0.19. Rosa [22] measured the Stribeck curve and tested the influence of particle size. The volume concentration of the particle was very low, and it was set properly to 1 vol. %. The results indicate that the COF is much lower for MR fluid with 0.4 μm particles than at MR fluid with 1.3 and 2 μm, which have almost identical COF. Zhang [23] measured COF for different particle volume content in commercial MR fluid from Lord company. The COF were almost identical (ca. 0.35) for 22, 32, and 40 vol.% of CI particles. Leung [24] used, for wear and COF measurement, block-on-ring geometry equipped with a stirring mechanism. He used a commercial MR fluid MRF-132DG from LORD as a source of CI particles and substituted the original carrier fluid with similar viscosity as an original carrier fluid and with four times higher viscosity. The measured COF of suspensions with higher viscosity of carrier fluid was ca. 0.065 and it was almost identical for high concentration of CI particles. For lower concentration it was significantly lower. The suspension with low viscosity of carrier fluid had identical COF for all concentrations (ca. 0.08). It seems that the carrier fluid with higher viscosity can form sufficient lubricating film, which can ensure the separation of both surfaces in contact into such a level that the particles can move in the contact more freely.

The intention of detailed state-of-art synthesis was the comparison of the results from various studies. Only tests with MRF-132DG from Lord company or fluids with similar composition, performed in modes pin-on-disc, block-on-ring or block-on-ball on three plates were evaluated. From these data, the viscosity of carrier fluids, the contact pressure, and the sliding speed of surfaces in contact from several papers can be calculated or estimated. In some cases, the sliding speed was just estimated from RPM. The viscosity for a certain temperature during measurement was also estimated. The contact pressure was calculated as the ratio of the normal force to the contact area calculated from the used geometry according to Hertz. The estimated (red) and calculated (blue) values are presented in Table 1. In an MR damper, two important steel–steel tribological contacts are operating in the MR fluid, namely, the piston rod guide and piston guide (no effect of a magnetic field). Considering a piston velocity in the range of 0.05 to 2 m/s, a lateral force in the range of 50 to 1000 N, a piston diameter of 32 mm, and a piston rod diameter of 12 mm, the range of a Hersey number between 1.E-11 and 1.E-09 for the piston rod guide and piston guide was determined. It can be seen that there are relatively few studies in the usual working condition of an MR damper.

The main goal of the paper is to determine the COF of MR fluids in a large range of a Hersey numbers and to focus mainly on the working conditions in the MR damper, which is essential for practical applications. The secondary aims are to describe the effects of particles on COF in the largest possible range of Hersey number, the effect of particle concentration, and the effect of temperature.

## 2. Materials and Methods

### 2.1. Measurement Method of COF

In this study, a rotating friction tester was used to carry out all experiments. Tests were performed on the tribometer Bruker ZP-UMT TriboLab (ZP-44957) (Figure 1a) with a pin-on-disc configuration (Figure 1b). This tester uses a rotary module to drive the disc sample fixed at the lower part. The pin was used during the measurement stationary with the possibility to set the radial and vertical position to the disc. The pin was fixed by an upper holder to the normal and radial force sensor. The measured MR fluid or oil was applied to the plate surface with a volume of 12 mL, which ensured the fully flooded contact. Experimental data, such as the friction force (F), temperature, and normal force (N) are measured using sensors. The COF is calculated using the following equation:(1)$$\mathrm{COF}=\frac{F}{P}$$
where COF is the kinetic coefficient of friction, F is the nominal measured friction force during sliding, and P is the applied load (normal force). The normal force is ensured by an upper linear drive and the permanent pressure of the pin to the disc is ensured by the springy metal strips Figure 1b. A hardened ball bearing with a diameter of 6.35 mm was used as a pin and a hardened tool steel as a disc. The disc had to be gently ground and polished up to less than Ra 0.025 µm to avoid the seizure at higher speed (Figure 2a). Generally, higher disc roughness resulted in seizure, especially at higher sliding speeds. However even polishing the disc did not fully prevent the contact from seizure (Figure 2b) because of the flexible mount of the pin, which started to vibrate at high turning speeds. For this reason, the maximal rotary speed had to be set on ca. 2400 rev/min. The corresponding sliding speed was insufficient to achieve the required values of the Hersey number. Therefore, to measure the Stribeck curve with this regime of lubrication, a normal force cannot stay constant for all measuring steps, and it had to be gradually reduced when the maximum permitted rotary speed was reached.

Thus, the normal force was gradually decreased from 20 to 1 N. The revolution of the rotary test module was set according to the required sliding speed and actual radius of the pin on the disc sample. The sliding speed was changed from 5 to 4000 mm/s in 20 steps, which approximately corresponds to 2400 rpm for maximal sliding speed; see Table 2. The initialization procedure was performed for 120 s at sliding speed 125 mm/s and 60 s at sliding speed 1500 mm/s to neglect the effect of worn surfaces on measuring COF.

During the procedure of the measurement of one Stribeck curve, the hardened ball as a pin was twice changed to avoid contact seizure because of the escalated pin wear at a high rotational speed. The repeating of measurements was carried out for obtaining three Stribeck curves for one sample because of the higher statistical significance of data. An error bar was created from these data. The new radius, new pin, and new tested fluid were used for each repeating.

The measurement was carried out for temperatures of 40 and 80 °C using a temperature chamber by Bruker. The delivered software with a tribometer cannot compile a Stribeck curve including the sliding speed, contact pressure, and dynamic viscosity on the x-axis. This software works only with revolutions of rotary drive, which is misleading when using a different radius of the pin. Therefore, the new script in Matlab for data evaluation was created. The one measurement (one repeating) of the Stribeck curve consists of 20 steps, as mentioned above when each step corresponds to one point on the Stribeck curve. For each step, the 16 changes in sense of the rotation were set for better flooding of the contact. Sometimes, the measured data included several types of instabilities that were solved by this script. Viscosity is an important input for calculation of Hersey number. The suspension viscosity (MR fluid) is dependent on the volume fraction of particles and carrier fluid viscosity. However, the amount of particles in contact is not known because some particles are probably excluded from the contact zone [30]. Therefore, the authors decided to use for calculation of Hersey number carrier fluid viscosity of MRF. The script calculates a relation among relative sliding speed of surfaces in contact *v*, contact pressure *p_z_,* and dynamic viscosity of carrier fluid *η* and draws it on the x-axis (Hersey number). This relation is given by (2):(2)$$axis\;value\;=\frac{\eta \;\cdot \;v\;}{p_{z}}$$

### 2.2. Testing Samples

Commercial MR fluids supplied by the LORD companyoration (MRF-122EG, MRF-132DG, and MRF-140CG) were chosen as the MR fluid samples. The properties from LORD Technical Datasheets are stated in Table 3.

The Lord MR fluids were compared with base oil from the group of synthetic liquid hydrocarbons (poly-alpha-olefin). The tested MR fluids differed in iron particle concentration from 22 to 40 vol. %. The commonly accepted rheological model of MR fluid is the Bingham model:(3)$$\tau =\eta_{0}\dot{\gamma }+\tau_{0}\left(H\right)sign\left(\dot{\gamma }\right)\;\;at\;\;\left|\tau \right|\geq \left|\tau_{0}\left(H\right)\right|,\dot{\gamma }=0\;\mathrm{at}\;\left|\tau \right|\leq \left|\tau_{0}\left(H\right)\right|,$$
where τ is shear stress, $\eta_{0}$ is plastic viscosity, $\tau_{0}$ is yield stress, H is magnetic flux intensity in MR fluid, and $\dot{\gamma }$ is shear strain rate. In the case of experiments, magnetic flux intensity is zero (H = 0 A/m). MR fluid in the non-activated state (H = 0 A/m) exhibits yield stress in tens of Pa. However, the MR fluid supplier usually states the Newtonian behavior of MR fluid and provides the value of viscosity at high shear rates. The particle size and distribution were measured by a scanning electron microscope FEG SEM ZEISS Ultra Plus and analyzed by script using tools for picture analysis in Matlab (R2018b). The particles were spherical and the average size (diameter approximately 2.1 µm) and distribution according to Q3 were identical for all MR fluids (Figure 3).

## 3. Results and Discussion

### 3.1. COF Comparison of Oils and MR Fluid

Figure 4 compares the Stribeck curves of measured PAO and MRF-132DG for a temperature of 40 °C. The results in Figure 4 indicate very high repeatability of the measurement; see error bars. Only at the boundary lubrication regime was the variance of values slightly higher. However, that can be understandable because in this area the solid surfaces come into contact and this event has a stochastic character. It can be seen that the boundary, mixed, elastohydrodynamic (EHD), and hydrodynamic regimes of lubrication were measured; see Figure 4. However, the EHD regime (the area with the lowest COF) at MR fluid came on at significantly higher COF than PAO, and then it rapidly rose. The EHD is defined as an area where the lubrication film has the same order of magnitude as the surface roughness. In this area, the iron particles of MR fluid probably work as a mediator at the interaction of both solid surfaces. The rapid increase of COF at the hydrodynamic regime at MR fluids can be explained by the presence of iron particles, which can form a higher lubricating wedge under the pin. The non-Newtonian rheological behavior of MR fluid may also affect the rapid increase of COF. The results indicate that the area with high relative sliding speed cannot be recommended as an operating point for MR fluid, which is the main contrast to common oils. Generally, for common oils, the hydrodynamic lubrication regime is desirable as an operating area. The hydrodynamic regime is defined as a regime fully separated from contacting surfaces by lubrication film and therefore the wear and COF are the lowest in this area. The measurements showed that the HD regime is also recommended for MR devices, but only in the narrow band up to sliding speed corresponding to COF of ca. 0.14. The measured value of COF in a mixed regime corresponds to the data published by Sohn [25] (COF 0.1–0.12) which was tested in the same configuration.

### 3.2. Particle Concentration Effect

It is surprising that the COF in the boundary lubrication regime is almost identical for all fluids, despite the presence of iron particles in MR fluid. Probably, only the viscosity of the carrier fluid and the friction surfaces affect the COF. It can be also deduced that the concentration of iron particles was already high enough for the fluid MRF-122EG (22 vol. %) to fully fill the lubrication gap between surfaces. Many studies suggest that the concentration above 10 vol. % of iron particles does not influence COF and wear [20]. In the hydrodynamic regime, the situation is different. The effect of particle concentration on COF is observable; see Figure 5. The higher the particle concentration, the higher the COF. The viscosity of the suspension (MR fluid) probably has a significant effect in the HD regime. Similar conclusions are presented in the paper [27].

### 3.3. Temperature Effect

Figure 6 shows the comparison between 40 and 80 °C, which can be considered as an operating point for linear actuators such as dampers. The viscosity was kept the same for both temperatures (0.01 Pa.s). No significant temperature effect was observed for MR fluids 122 and 140. For MR fluid 132, there is a slight difference in the minimum COF value. However, the PAO was significantly affected by temperature. A significant increase in COF over the entire Hersey number range was observed at 80 °C; see Figure 6d.

## 4. Conclusions

In this work, the COF measurements for three commercial MR fluids with 22–40 vol. % of CI particles from LORD company oration were done. These fluids were compared with synthetic liquid hydrocarbon PAO4, which corresponds to the base oil of MR fluids. The properties of all the fluids were measured at temperatures 40 and 80 °C. The results indicate that in the range of boundary lubrication, all the fluids exhibit similar COF, despite iron particles in the MR fluids. In the EHD area (transition between mix and hydrodynamic lubrication), where the COF is generally the lowest, the MR fluids exhibit slightly higher values of COF than PAO. The biggest difference can be observed in the hydrodynamic regime. The COF for PAO rises with a much gentler slope than MR fluids. The effect of particle concentration on COF is observable. The temperature effect on the COF of MRF is insignificant.

The results can be also used as a guide for developers of MR devices. The important result is the specification of a suitable operating range. Considering low friction and total separation of contact surfaces by lubrication film, the hydrodynamic lubrication area seems to be suitable. On the contrary, the contact flooded by MR fluid should not be operated at higher Hersey numbers, because the COF rises very steeply for specific numbers, higher than the EHD lubrication regime.

A comparison of COF measured by authors mentioned in the introduction (see Figure 7) shows quite a large variance, although only measurements with MRF132DG or fluids with similar particle concentrations are included in this figure. The majority of the values were measured in boundary and mixed regimes, where the COF is very dependent on contact surface roughness. Only a few values were measured in the hydrodynamic regime. In the HD regime, the surface roughness and geometry should have a much smaller impact on measured COF; therefore, a smaller variance can be expected. This work provided information about the COF in the usual working area of the MR damper (HD regime).

## Figures and Tables

**Figure 1 materials-13-04670-f001:**
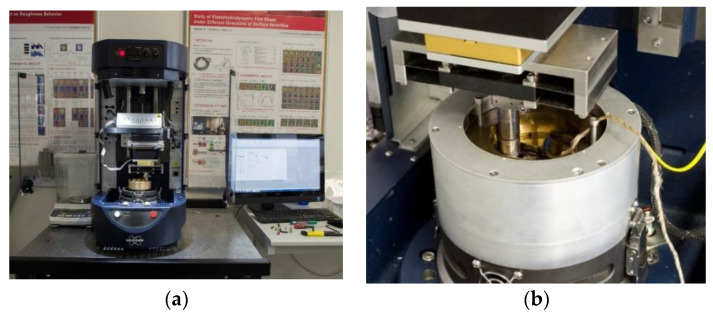
(**a**) Tribometer UMT Bruker; (**b**) pin-on-disc configuration with the heating module.

**Figure 2 materials-13-04670-f002:**
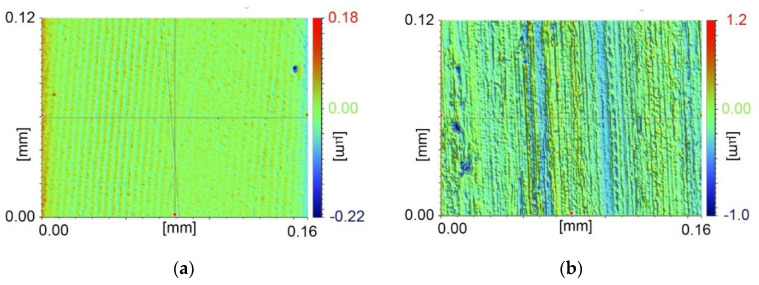
(**a**) Polished disc—Ra 0.014 µm; (**b**) track roughness after test—Ra 0.110 µm.

**Figure 3 materials-13-04670-f003:**
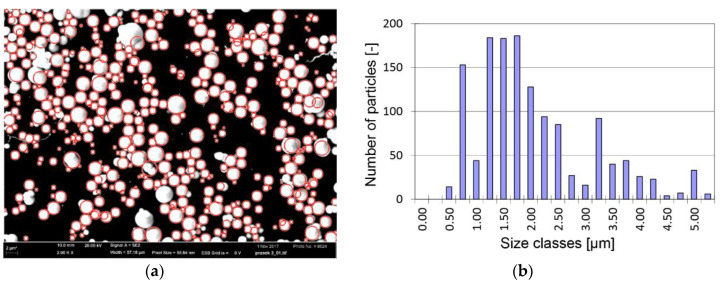
MRF-122EG (**a**) SEM picture of centrifuging particles; (**b**) histogram for 1399 detected particles.

**Figure 4 materials-13-04670-f004:**
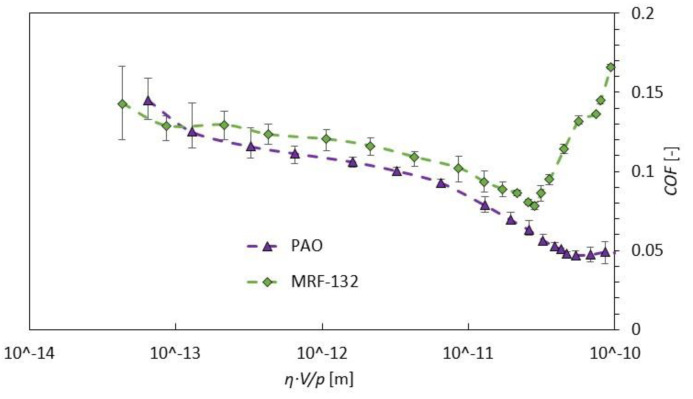
Stribeck curves comparison of PAO 4 and magnetorheological fluid MRF-132DG at 40 °C.

**Figure 5 materials-13-04670-f005:**
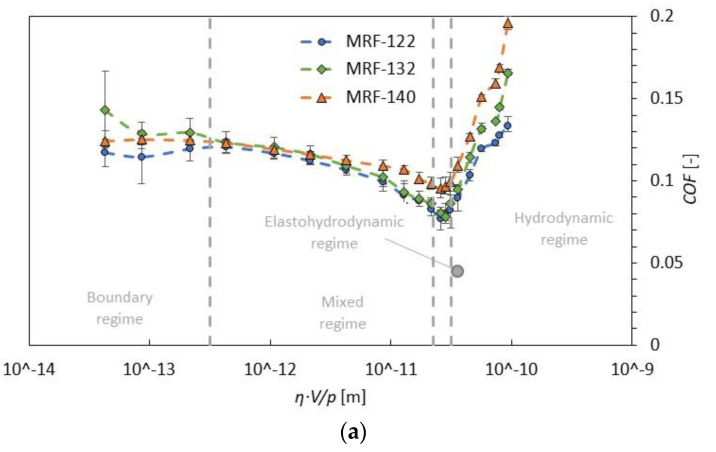
Stribeck curves of (**a**) MR fluids MRF-122EG, MRF-132DG, and MRF-140CG at 40 °C; (**b**) detail on HD regime.

**Figure 6 materials-13-04670-f006:**
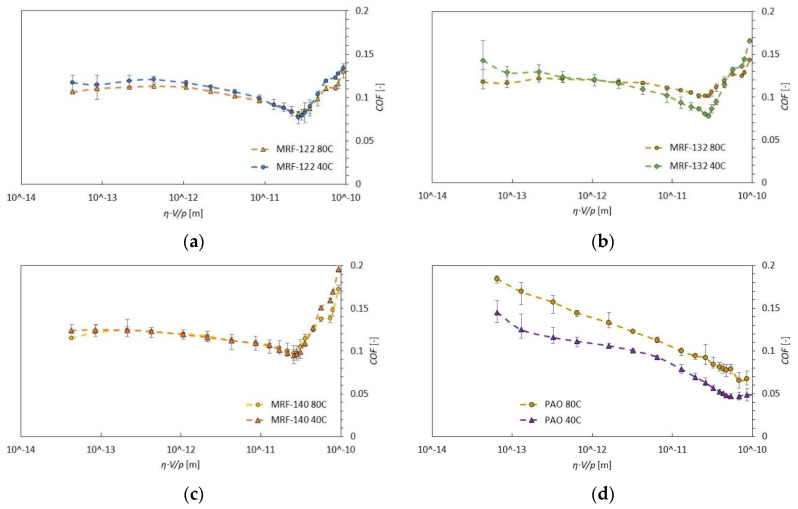
Comparison of individual measured fluid at temperatures 40 and 80 °C: (**a**) MRF-122; (**b**) MRF-132; (**c**) MRF-140; (**d**) PAO.

**Figure 7 materials-13-04670-f007:**
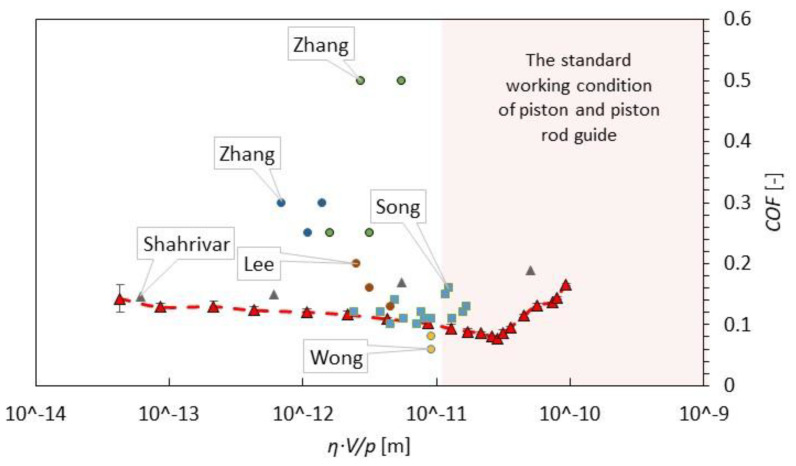
Comparison of results of COF from our study (red line) and results from previous works of other authors for MR fluids MRF-132DG or similar MR fluid.

**Table 1 materials-13-04670-t001:** Comparison of results from published studies (red: estimation value, blue: calculation) [20,21,23,25,26,27,28,29].

Author	Geometry	Relative Speed [mm/s]	Normal Force [N]	Contact Pair	Viscosity [mPa.s]	COF [-]	Contact Pressure [GPa]
Sohn	Block on ring	950	40	S45C/S45C	14.2	0.8	1.433
Song	pin on disc	780	50	brass/brass	14.2	0.13	0.012
Song	pin on disc	780	20	steel/steel	13.6	0.12	1.367
		780	50	steel/steel	13.6	0.11	1.855
		780	100	steel/steel	13.6	0.1	2.337
		780	20	brass/brass	13.6	0.16	0.862
		780	100	brass/brass	13.6	0.1	1.474
		780	20	alu/alu	13.6	0.12	0.673
		780	50	alu/alu	13.6	0.15	0.913
		780	100	alu/alu	13.6	0.11	1.150
		330	50	steel/steel	13.6	0.12	1.855
		1120	50	steel/steel	13.6	0.11	1.855
		330	50	brass/brass	13.6	0.12	1.170
		1120	50	brass/brass	13.6	0.11	1.170
		330	50	alu/alu	13.6	0.14	0.913
		1120	50	alu/alu	13.6	0.13	0.913
Wong	block on ring	1623	292	steel/steel	15.1	0.06	2.658
		1623	292	steel/steel	15.1	0.08	26.585
Zhang	pin on disk	31.25	1	alu/alu	13.6	0.5	0.156
		31.25	5	alu/alu	13.6	0.25	0.267
		62.5	1	alu/alu	13.6	0.5	0.156
		62.5	5	alu/alu	13.6	0.25	0.267
Zhang	pin on disk	31.25	10	steel/steel	15.1	0.3	0.683
		31.25	10	alu/alu	15.1	0.3	0.336
		31.25	10	brass/brass	15.1	0.25	0.336
Lee	pin on disc	390	50	steel/steel	15.1	0.16	1.855
		390	100	steel/steel	15.1	0.2	2.337
		560	50	steel/steel	15.1	0.13	1.855
Shahrivar	ball-on-three plates	0.06	10	steel/steel	13.6	0.15	0.224
		0.1	10	steel/steel	13.6	0.14	0.224
		1	10	steel/steel	13.6	0.145	0.224
		10	10	steel/steel	13.6	0.15	0.224
		100	11	steel/steel	13.6	0.17	0.224
		1000	12	steel/steel	13.6	0.19	0.269

**Table 2 materials-13-04670-t002:** Conditions for Stribeck curve measurement.

Step	F_z_ (N)	v (mm/s)	Step	F_z_ (N)	v (mm/s)
1	20	5	11	20	2500
2	20	10	12	20	3000
3	20	25	13	20	3300
4	20	50	14	15	3300
5	20	125	15	10	3300
6	20	250	16	5	3300
7	20	500	17	2.5	3300
8	20	1000	18	2	4000
9	20	1500	19	1.6	4000
10	20	2000	20	1	4000

**Table 3 materials-13-04670-t003:** Properties of MR fluid samples and PAO.

Parameter	MRF-122EG	MRF-132DG	MRF-140CG	PAO4
Viscosity at 40 °C (Pa.s)	0.042 ± 0.02	0.112 ± 0.02	0.28 ± 0.07	0.014
Density (g/cm^3^)	2.28–2.48	2.95–3.15	3.54–3.74	0.82
Solids content of weight (%)	72	80.98	85.44	0
Solids content of volume (%)	22	32	40	0

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
