# Peer review of "Stribeck Curve of Magnetorheological Fluid within Pin-on-Disc Configuration: An Experimental Investigation"

_materials, 2020, doi:10.3390/ma13204670_

Round 1
Reviewer 1 Report
The author discussed the stribeck curve study of magnetorheological fluid and compared it with PAO base oil. Some questions should be addressed before acceptance.
1. The author conducted the friction tests with changing speed and load in 20 steps. It is noted that the author started with high load and low speed, which corresponds to the low Hersey number (boundary lubrication regime). Did the wear affect further results?
2. In the stribeck curve plot, the x-axis Hersey number should be a dimensionless parameter. What does the [m] mean?
3. In the results and discussion session, the author briefly described the phenomenon and figures. The discussion and mechanism were necessary.
4. Why was the boundary lubrication behavior similar for all fluids? Did the author conduct any characterization on worn surfaces?
5. In Conclusions, “In EHD area…” The EHD area should be the transition between the mix regime and HD lubrication.
Reviewer 2 Report
This paper investigates the coefficient of friction of a magnetorheological fluid. The effects of particle concentration and temperature were studied. The paper is generally well written and the presentation is clear. The scientific contribution of this paper can be more clearly articulated. The authors can discuss the effects of different regimes in more detail and provide more explanation to improve the depth of this study.
Specific comments:
-Table 1 is a bit difficult to read
-Definition of different regimes can be more clearly defined (mixed, EHD, HD…). It would be great to label them in those relevant figures.
-Some grammatical issues can be fixed with a thorough proofread.
-Scale bar in Fig. 4 can be made clearer.
-In L158, should that be Figure 4?
-Fig. 1 and Fig. 8 are a bit redundant. It might not be necessary to show both.
Reviewer 3 Report
Overall Recommendation: Accepted after minor revision
This paper reports an experimental study of magnetorheological fluids under different lubrication regimes. Their experiments show that particle concentration plays an important role in the hydrodynamic regime. The manuscript is well written, it is highly logical in the way that the authors present their results, and they compare with many already reported data from previous experiments, showing the relevance of their results. Therefore, the results appear to be worthy of publication.
It is recommended a minor change in the format of the figures: Zoom in the plots, such that the figure box tightly encloses the data. In this way, you will remove a lot of unused space in every plot.
Reviewer 4 Report
This article examines the role of COF of the magnetorheological fluid within the specific configuration by several experimental trials. The topic seems to be important and significant to the related field but the paper lacks of the theoretical explanation of the phenomenon. Furthermore, even if the authors heavily depends on the experimental results, their results seem not to be reliable. For this point of view, this paper can not be considered to be published in the present form unless the following comments are addressed.
- (Major) Although the paper considers on the effect of COF of MR, which can be verified by experimental results, the authors should include the theoretical background of the behavior of MR fluid within the irregular shaped domain. This referee recommends to survey on the theoretical results on the similar configuration of the article and introduce them in the paper.
- (Major) Experimental results in section 3 seem to be ambiguous for supporting the argument of the paper, which must be the main part of this paper. The experimental trials have to be accurate and reliable by choosing proper comparable experiments. This needs to be fixed.
- (Minor) There are several typos and wrong grammars. Read thoroughly and fix them.
Round 2
Reviewer 4 Report
The comments from this referee have been addressed properly. This article can be accepted in this form.